# Pass-Efficient Unsupervised Feature Selection

**Crystal Maung**
Department of Computer Science
The University of Texas at Dallas
Crystal.Maung@gmail.com

**Haim Schweitzer**
Department of Computer Science
The University of Texas at Dallas
HSchweitzer@utdallas.edu

## Abstract

The goal of unsupervised feature selection is to identify a small number of important features that can represent the data. We propose a new algorithm, a modification of the classical pivoted QR algorithm of Businger and Golub, that requires a small number of passes over the data. The improvements are based on two ideas: keeping track of multiple features in each pass, and skipping calculations that can be shown not to affect the final selection. Our algorithm selects the exact same features as the classical pivoted QR algorithm, and has the same favorable numerical stability. We describe experiments on real-world datasets which sometimes show improvements of *several orders of magnitude* over the classical algorithm. These results appear to be competitive with recently proposed randomized algorithms in terms of pass efficiency and run time. On the other hand, the randomized algorithms may produce more accurate features, at the cost of small probability of failure.

## 1 Introduction

Work on unsupervised feature selection has received considerable attention. See, e.g., [1, 2, 3, 4, 5, 6, 7, 8] . In numerical linear algebra unsupervised feature selection is known as the *column subset selection problem*, where one attempts to identify a small subset of matrix columns that can approximate the entire column space of the matrix. See, e.g., [9, Chapter 12]. The distinction between supervised and unsupervised feature selection is as follows. In the supervised case one is given labeled objects as training data and features are selected to help predict that label; in the unsupervised case nothing is known about the labels.

We describe an improvement to the classical Businger and Golub pivoted QR algorithm [9, 10]. We refer to the original algorithm as the **QRP**, and to our improved algorithm as the **IQRP**. The QRP selects features one by one, using $k$ passes in order to select $k$ features. In each pass the selected feature is the one that is the hardest to approximate by the previously selected features. We achieve improvements to the algorithm run time and pass efficiency without affecting the selection and the excellent numerical stability of the original algorithm. Our algorithm is deterministic, and runs in a small number of passes over the data. It is based on the following two ideas:

1. In each pass we identify multiple features that are hard to approximate with the previously selected features. A second selection step among these features uses an upper bound on unselected features that enables identifying multiple features that are guaranteed to have been selected by the QRP. See Section 4 for details.
2. Since the error of approximating a feature can only decrease when additional features are added to the selection, there is no need to evaluate candidates with error that is already "too small". This allows a significant reduction in the number of candidate features that need to be considered in each pass. See Section 4 for details.

# 2 Algorithms for unsupervised feature selection

The algorithms that we consider take as input large matrices of numeric values. We denote by $m$ the number of rows, by $n$ the number of columns (features), and by $k$ the number of features to be selected. Criteria for evaluating algorithms include their run time and memory requirements, the number of passes over the data, and the algorithm accuracy. The accuracy is a measure of the error of approximating the entire data matrix as a linear combination of the selection. We review some classical and recent algorithms for unsupervised feature selection.

## 2.1 Related work in numerical linear algebra

**Businger and Golub QRP** was established by Businger and Golub [9, 10]. We discuss it in detail in Section 3. It requires $k$ passes for selecting $k$ features, and its run time is $4kmn - 2k^2(m+n) + 4k^3/3$. A recent study [11] compares experimentally the accuracy of the QRP as a feature selection algorithm to some recently proposed state-of-the-art algorithms. Even though the accuracy of the QRP is somewhat below the other algorithms, the results are quite similar. (The only exception was the performance on the Kahan matrix, where the QRP was much less accurate.)

**Gu and Eisenstat** algorithm [1] was considered the most accurate prior to the work on randomized algorithms that had started with [12]. It computes an initial selection (typically by using the QRP), and then repeatedly swaps selected columns with unselected column. The swapping is done so that the product of singular values of the matrix formed by the selected columns is increased with each swapping. The algorithm requires random access memory, and it is not clear how to implement it by a series of passes over the data. Its run time is $O(m^2 n)$.

## 2.2 Randomized algorithms

Randomized algorithms come with a small probability of failure, but otherwise appear to be more accurate than the classical deterministic algorithms. Frieze et al [12, 13] have proposed a randomized algorithm that requires only two passes over the data. This assumes that the norms of all matrix columns are known in advance, and guarantees only an additive approximation error. We discuss the run time and the accuracy of several generalizations that followed their studies.

**Volume sampling** Deshpande et al [14] have studied a randomized algorithm that samples $k$-tuples of columns with probability proportional to their "volume". The volume is the square of the product of the singular values of the submatrix formed by these columns. They show that this sampling scheme gives rise to a randomized algorithm that computes the best possible solution in the Frobenius norm. They describe an efficient $O(kmn)$ randomized algorithm that can be implemented in $k$ passes and approximates this sampling scheme. These results were improved (in terms of accuracy) in [15], by computing the exact volume sampling. The resulting algorithm is slower but much more accurate. Further improvements to the speed of volume sampling in [6] have reduced the run time complexity to $O(km^2 n)$. As shown in [15, 6], this optimal (in terms of accuracy) algorithm can also be derandomized, with a deterministic run time of $O(km^3 n)$.

**Leverage sampling** The idea behind leverage sampling is to randomly select features with probability proportional to their "leverage". Leverage values are norms of the rows of the $n \times k$ right eigenvector matrix in the truncated SVD expansion of the data matrix. See [16, 2]. In particular, the "two stage" algorithm described in [2] requires only 2 passes if the leverage values are known. Its run time is dominated by the calculation of the leverage values. To the best of our knowledge the currently best algorithms for estimating leverage values are randomized [17, 18]. One run takes 2 passes and $O(mn \log n + m^3)$ time. This is dominated by the $mn$ term, and [18] show that it can be further reduced to the number of nonzero values. We note that these algorithms do not compute reliable leverage in 2 passes, since they may fail at a relatively high (e.g., 1/3) probability. As stated in [18] "the success probability can be amplified by independent repetition and taking the coordinate-wise median". Therefore, accurate estimates of leverage can be computed in constant number of passes. But the constant would be larger than 2.

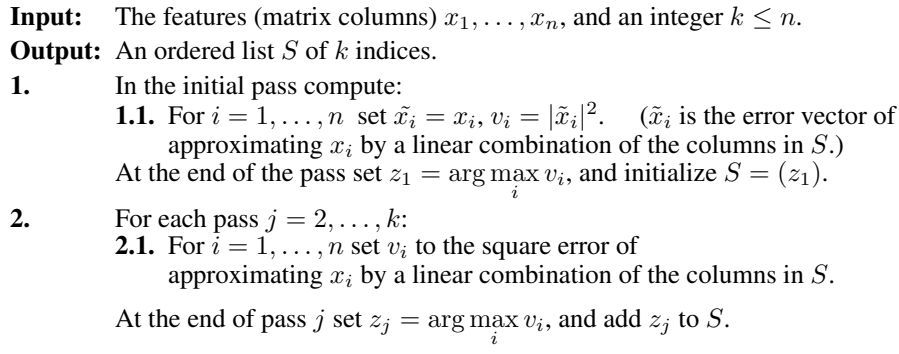

| **Input:** | The features (matrix columns) $x_1, \ldots, x_n$, and an integer $k \le n$. |
| --- | --- |

**Output:** An ordered list $S$ of $k$ indices.

**1.** In the initial pass compute:
  **1.1.** For $i = 1, \ldots, n$ set $\tilde{x}_i = x_i$, $v_i = |\tilde{x}_i|^2$.    ($\tilde{x}_i$ is the error vector of approximating $x_i$ by a linear combination of the columns in $S$.)
  At the end of the pass set $z_1 = \arg\max_i v_i$, and initialize $S = (z_1)$.

**2.** For each pass $j = 2, \ldots, k$:
  **2.1.** For $i = 1, \ldots, n$ set $v_i$ to the square error of approximating $x_i$ by a linear combination of the columns in $S$.

  At the end of pass $j$ set $z_j = \arg\max_i v_i$, and add $z_j$ to $S$.

Figure 1: The main steps of the QRP algorithm.

## 2.3 Randomized ID

In a recent survey [19] Halko et.al. describe how to compute QR factorization using their randomized Interpolative Decomposition. Their approach produces an accurate $Q$ as a basis of the data matrix column space. They propose an efficient "row extraction" method for computing $R$, that works when $k$, the desired rank, is similar to the rank of the data matrix. Otherwise the row extraction introduces unacceptable inaccuracies, which led Halko et.al to recommend using an alternative $O(kmn)$ technique in such cases.

## 2.4 Our result, the complexity of the IQRP

The savings that the IQRP achieves depend on the data. The algorithm takes as input an integer value $l$, the length of a temporary buffer. As explained in Section 4 our implementation requires temporary storage of $l + 1$, which takes $(l + 1)m$ floats. The following values depend on the data: the number of passes $p$, the number of IO-passes $q$ (explained below), and a unit cost of orthogonalization $c$ (see Section 4.3).

In terms of $l$ and $c$ the run time is $2mn + 4mnc + 4mlk$. Our experiments show that for typical datasets the value of $c$ is below $k$. For $l \approx k$ our experiments show that the number of passes is typically much smaller than $k$. The number of passes is even smaller if one considers *IO-passes*. To explain what we mean by *IO-passes* consider as an example a situation where the algorithm runs three passes over the data. In the first pass all $n$ features are being accessed. In the second, only two features are being accessed. In the third, only one feature is being accessed. In this case we take the number of IO-passes to be $q = 1 + \frac{3}{n}$. We believe that $q$ is a relevant measure of the algorithm pass complexity when skipping is cheap, so that the cost of a pass over the data is the amount of data that needs to be read.

## 3 The Businger Golub algorithm (QRP)

In this section we describe the QRP [9, 10] which forms the basis to the IQRP. The main steps are described in Figure 1. There are two standard implementations for Step 2.1 in Figure 1. The first is by means of the "Modified Gram-Schmidt" (e.g., [9]), and the second is by Householder orthogonalization (e.g., [9]). Both methods require approximately the same number of flops, but error analysis (see [9]) shows that the Householder approach is significantly more stable.

### 3.1 Memory-efficient implementations

The implementations shown in Figure 2 update the memory where the matrix $A$ is stored. Specifically, $A$ is overwritten by the $R$ component of the QR factorization. Since we are not interested in $R$, overwriting $A$ may not be acceptable. The procedure shown in Figure 3 does not overwrite $A$, but it is more costly. The flops count is dominated by Steps 1 and 2, which cost at most $4(j - 1)mn$ at pass $j$. Summing up for $j = 1, \ldots, k$ this gives a total flops count of approximately $2k^2 mn$ flops.

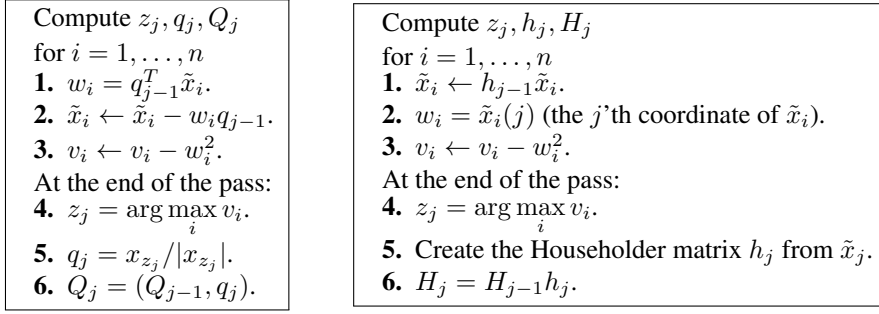

Modified Gram-Schmidt          Householder orthogonalization

Figure 2: Standard implementations of Step 2.1 of the QRP

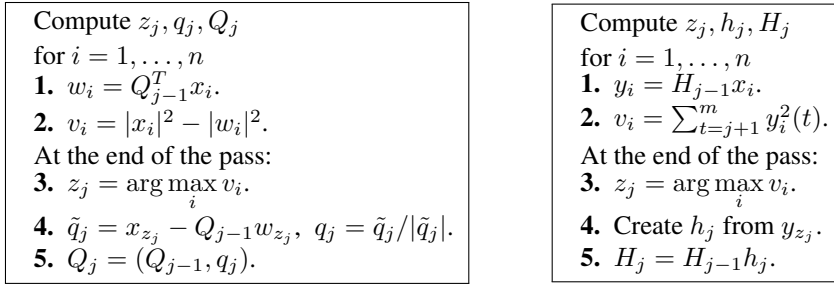

Modified Gram-Schmidt          Householder orthogonalization

Figure 3: Memory-efficient implementations of Step 2.1 of the QRP

## 4 The IQRP algorithm

In this section we describe our main result: the improved QRP. The algorithm maintains three ordered lists of columns: **The list $F$** is the input list containing all columns. **The list $S$** contains columns that have already been selected. **The list $L$** is of size $l$, where $l$ is a user defined parameter.

For each column $x_i$ in $F$ the algorithm maintains an integer value $r_i$ and a real value $v_i$. These values can be kept in core or a secondary memory. They are defined as follows:

$$r_i \leq |S|, \quad v_i = v_i(r_i) = \|x_i - Q_{r_i} Q_{r_i}^T x_i\|^2 \tag{1}$$

where $Q_{r_i} = (q_1, \ldots, q_{r_i})$ is an orthonormal basis to the first $r_i$ columns in $S$. Thus, $v_i(r_i)$ is the (squared) error of approximating $x_i$ with the first $r_i$ columns in $S$. In each pass the algorithm identifies the $l$ candidate columns $x_i$ corresponding to the $l$ largest values of $v_i(|S|)$. That is, the $v_i$ values are computed as the error of predicting each candidate by *all* columns currently in $S$. The identified $l$ columns with the largest $v_i(|S|)$ are stored in the list $L$. In addition, the value of the $l+1$'th largest $v_i(|S|)$ is kept as the constant $B_F$. Thus, after a pass is terminated the following condition holds:

$$v_\alpha(r_\alpha) \leq B_F \quad \text{for all } x_\alpha \in F \setminus L. \tag{2}$$

The list $L$ and the value $B_F$ can be calculated in one pass using a binary heap data structure, with the cost of at most $n \log(l+1)$ comparisons. See [20, Chapter 9]. The main steps of the algorithm are described in Figure 4.

**Details of Steps 2.0, 2.1 of the IQRP.** The threshold value $T$ is defined by:

$$T = \begin{cases} -\infty & \text{if the heap is not full.} \\ \text{top of the heap} & \text{if the heap is full.} \end{cases} \tag{3}$$

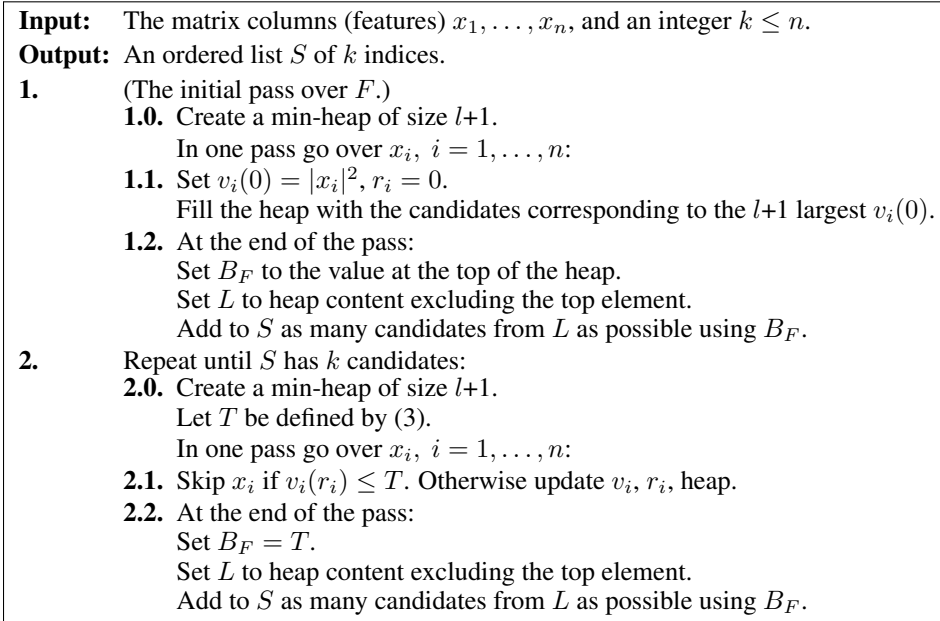

**Input:** The matrix columns (features) $x_1, \ldots, x_n$, and an integer $k \leq n$.
**Output:** An ordered list $S$ of $k$ indices.
**1.** (The initial pass over $F$.)
    **1.0.** Create a min-heap of size $l+1$.
        In one pass go over $x_i$, $i = 1, \ldots, n$:
    **1.1.** Set $v_i(0) = |x_i|^2$, $r_i = 0$.
        Fill the heap with the candidates corresponding to the $l+1$ largest $v_i(0)$.
    **1.2.** At the end of the pass:
        Set $B_F$ to the value at the top of the heap.
        Set $L$ to heap content excluding the top element.
        Add to $S$ as many candidates from $L$ as possible using $B_F$.
**2.** Repeat until $S$ has $k$ candidates:
    **2.0.** Create a min-heap of size $l+1$.
        Let $T$ be defined by (3).
        In one pass go over $x_i$, $i = 1, \ldots, n$:
    **2.1.** Skip $x_i$ if $v_i(r_i) \leq T$. Otherwise update $v_i, r_i$, heap.
    **2.2.** At the end of the pass:
        Set $B_F = T$.
        Set $L$ to heap content excluding the top element.
        Add to $S$ as many candidates from $L$ as possible using $B_F$.

Figure 4: The main steps of the IQRP algorithm.

Thus, when the heap is full, $T$ is the value of $v$ associated with the $l+1$'th largest candidate encountered so far. The details of Step 2.1 are shown in Figure 5. Step A.2.2.1 can be computed using either Gram-Schmidt or Householder, as shown in Figures 3 and 4.

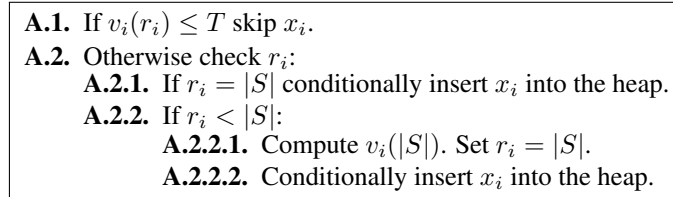

**A.1.** If $v_i(r_i) \leq T$ skip $x_i$.
**A.2.** Otherwise check $r_i$:
    **A.2.1.** If $r_i = |S|$ conditionally insert $x_i$ into the heap.
    **A.2.2.** If $r_i < |S|$:
        **A.2.2.1.** Compute $v_i(|S|)$. Set $r_i = |S|$.
        **A.2.2.2.** Conditionally insert $x_i$ into the heap.

Figure 5: Details of Step 2.1

**Details of Steps 1.2 and 2.2 of the IQRP.** Here we are given the list $L$ and the value of $B_F$ satisfying (2). To move candidates from $L$ to $S$ run the QRP on $L$ as long as the pivot value is above $B_F$. (The pivot value is the largest value of $v_i(|S|)$ in $L$.) The details are shown in Figure 6.

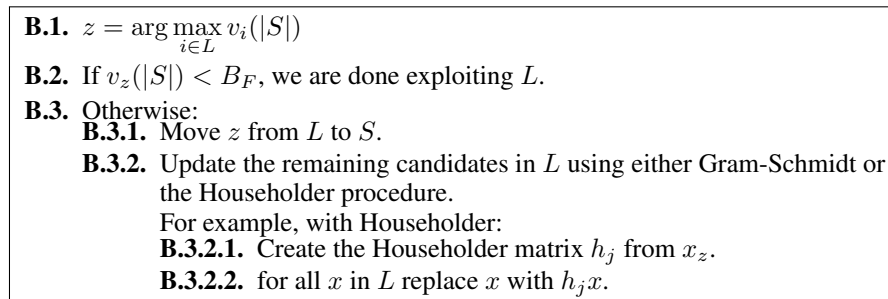

**B.1.** $z = \arg\max\limits_{i \in L} v_i(|S|)$
**B.2.** If $v_z(|S|) < B_F$, we are done exploiting $L$.
**B.3.** Otherwise:
    **B.3.1.** Move $z$ from $L$ to $S$.
    **B.3.2.** Update the remaining candidates in $L$ using either Gram-Schmidt or the Householder procedure.
        For example, with Householder:
        **B.3.2.1.** Create the Householder matrix $h_j$ from $x_z$.
        **B.3.2.2.** for all $x$ in $L$ replace $x$ with $h_j x$.

Figure 6: Details of Steps 1.2 and 2.2

## 4.1 Correctness

In this section we show that the IQRP computes the same selection as the QRP. The proof is by induction on $j$, the number of columns in $S$. For $j = 0$ the QRP selects $x_j$ with $v_j = |x_j|^2 = \max_i |x_i|^2$. The IQRP selects $v_j'$ as the largest among the $l$ largest values in $F$. Therefore $v_j' = \max_{x_i \in L} |x_i|^2 = \max_{x_i \in F} |x_i|^2 = v_j$. Now assume that for $j = |S|$ the QRP and the IQRP select the same columns in $S$ (this is the inductive assumption). Let $v_j(|S|)$ be the value of the $j$+1'th selection by the QRP, and let $v_j'(|S|)$ be the value of the $j$+1'th selection by the IQRP. We need to show that $v_j'(|S|) = v_j(|S|)$. The QRP selection of $j$ satisfies: $v_j(|S|) = \max_{x_i \in F} v_i(|S|)$. Observe that if $x_i \in L$ then $r_i = |S|$. (Initially $L$ is created from the heap elements that have $r_i = |S|$. Once $S$ is increased in Step B.3.1 the columns in $L$ are updated according to B.3.2 so that they all satisfy $r_i = |S|$.) The IQRP selection satisfies:

$$v_j'(|S|) = \max_{x_i \in L} v_i(|S|) \quad \text{and} \quad v_j'(|S|) \geq B_F. \tag{4}$$

Additionally for all $x_\alpha \in F \setminus L$:

$$B_F \geq v_\alpha(r_\alpha) \geq v_\alpha(|S|). \tag{5}$$

This follows from (2), the observation that $v_\alpha(r)$ is monotonically decreasing in $r$, and $r_\alpha \leq |S|$. Therefore, combining (4), and (5) we get

$$v_j'(|S|) = \max_{x_i \in F} v_i(|S|) = v_j(|S|).$$

This completes the proof by induction.

## 4.2 Termination

To see that the algorithm terminates it is enough to observe that at least one column is selected in each pass. The condition at Step B.2 in Figure 6 cannot hold at the first time in a new $L$. The value of $B_F$ is the $l$+1 largest $v_i(|S|)$, while the maximum at B.1 is among the $l$ largest $v_i(|S|)$.

## 4.3 Complexity

The formulas in this section describe the complexity of the IQRP in terms of the following:

| | | | |
|---|---|---|---|
| $n$ | the number of features (matrix columns) | $m$ | the number of objects (matrix rows) |
| $k$ | the number of selected features | $l$ | user provided parameter. $1 \leq l \leq n$ |
| $p$ | number of passes | $q$ | number of IO-passes |
| $c$ | a unit cost of orthogonalizing $F$ | | |

The value of $c$ depends on the implementation of Step A.2.2.1 in Figure 5. We write $c_{\text{memory}}$ for the value of $c$ in the memory-efficient implementation, and $c_{\text{flops}}$ for the faster implementation (in terms of flops). We use the following notation. At pass $j$ the number of selected columns is $k_j$, and the number of columns that were not skipped in Step 2.1 of the IQRP (same as Step A.1) is $n_j$.

The number of flops in the memory-efficient implementation can be shown to be

$$\text{flops}_{\text{memory}} = 2mn + 4mnc + 4mlk, \quad \text{where } c = \sum_{j=2}^{p} \frac{n_j}{n} \sum_{j'=1}^{j-1} k_{j'} \tag{6}$$

Observe that $c \leq k^2/2$, so that for $l < n$ the worst case behavior is the same as the memory-optimized QRP algorithm, which is $O(k^2 mn)$. We show in Section 5 that the typical run time is much faster. In particular, the dependency on $k$ appears to be linear and not quadratic.

For the faster implementation that overwrites the input it can be shown that:

$$\text{flops}_{\text{time}} = 2mn + 4m \sum_{i=1}^{n} \tilde{r}_i, \quad \text{where } \tilde{r}_i \text{ is the value of } r_i \text{ at termination.} \tag{7}$$

Since $\tilde{r}_i \leq k - 1$ it follows that $\text{flops}_{\text{time}} \leq 4kmn$. Thus, the worst case behavior is the same as the flops-efficient QRP algorithm.

Memory in the memory-efficient implementation requires $km$ in-core floats, and additional memory for the heap, that can be reused for the list $L$. Additional memory to store and manipulate $v_i, r_i$ for $i = 1, \ldots, n$ is roughly $2n$ floats. Observe that these memory locations are being accessed consecutively, and can be efficiently stored and manipulated out-of-core. The data itself, the matrix $A$, is stored out-of-core. When the method of Figure 3 is used in A.2.2.1, these matrix values are read-only.

IO-passes. We wish to distinguish between a pass where the entire data is accessed and a pass where most of the data is skipped. This suggests the following definition for the number of *IO-passes*: $q = \sum_{j=1}^{p} \frac{n_j}{n} = 1 + \sum_{j=2}^{p} \frac{n_j}{n}$.

Number of floating point comparisons. Testing for the skipping and manipulating the heap requires floating point comparisons. The number of comparisons is $n(p - 1 + (q - 1) \log_2(l + 1))$. This does not affect the asymptotic complexity since the number of flops is much larger.

## 5    Experimental results

We describe results on several commonly used datasets. **"Day1"**, with $m = 20,000$ and $n = 3,231,957$ is part of the "URL_reputation" collection at the UCI Repository. **"thrombin"**, with $m = 1,909$ and $n = 139,351$ is the data used in KDD Cup 2001. **"Amazon"**, with $m = 1,500$ and $n = 10,000$ is part of the "Amazon Commerce reviews set" and was obtained from the UCI Repository. **"gisette"**, with $m = 6,000$ and $n = 5,000$ was used in NIPS 2003 selection challenge.

**Measurements.**   We vary $k$, and report the following: **flops$_{\text{memory}}$, flops$_{\text{time}}$** are the ratios between the number of flops used by the IQRP and $kmn$, for the memory-efficient orthogonalization and the time-efficient orthogonalization. **# passes** is the number of passes needed to select $k$ features. **# IO-passes** is discussed in sections 2.4 and 4.3. It is the number of times that the entire data is read. Thus, the ratio between the number of IO-passes and the number of passes is the fraction of the data that was not skipped.

**Run time.**   The number of flops of the QRP is between $2kmn$ and $4kmn$. We describe experiments with the list size $l$ taken as $l = k$. For Day1 the number of flops beats the QRP by a factor of more than 100. For the other datasets the results are not as impressive. There are still significant savings for small and moderate values of $k$ (say up to $k = 600$), but for larger values the savings are smaller. Most interesting is the observation that the memory-efficient implementation of Step 2.1 is not much slower than the optimization for time. Recall that the memory-optimized QRP is $k$ times slower than the time-optimized QRP. In our experiments they differ by no more than a factor of 4.

**Number of passes.**   We describe experiments with the list size $l$ taken as $l = k$, and also with $l = 100$ regardless of the value of $k$. The QRP takes $k$ passes for selecting $k$ features. For the Day1 dataset we observed a reduction by a factor of between 50 to 250 in the number of passes. For IO-passes, the reduction goes up to a factor of almost 1000. Similar improvements are observed for the Amazon and the gisette datasets. For the thrombin it is slightly worse, typically a reduction by a factor of about 70. The number of IO-passes is always significantly below the number of passes, giving a reduction by factors up to 1000. For the recommended setting of $l = k$ we observed the following. In absolute terms the number of passes was below 10 for most of the data; the number of IO-passes was below 2 for most of the data.

## 6    Concluding remarks

This paper describes a new algorithm for unsupervised feature selection. Based on the experiments we recommend using the memory-efficient implementation and setting the parameter $l = k$. As explained earlier the algorithm maintains 2 numbers for each column, and these can also be kept in-core. This gives a $2(km + n)$ memory footprint.

Our experiments show that for typical datasets the number of passes is significantly smaller than $k$. In situations where memory can be skipped the notion of IO-passes may be more accurate than passes. IO-passes indicate the amount of data that was actually read and not skipped.

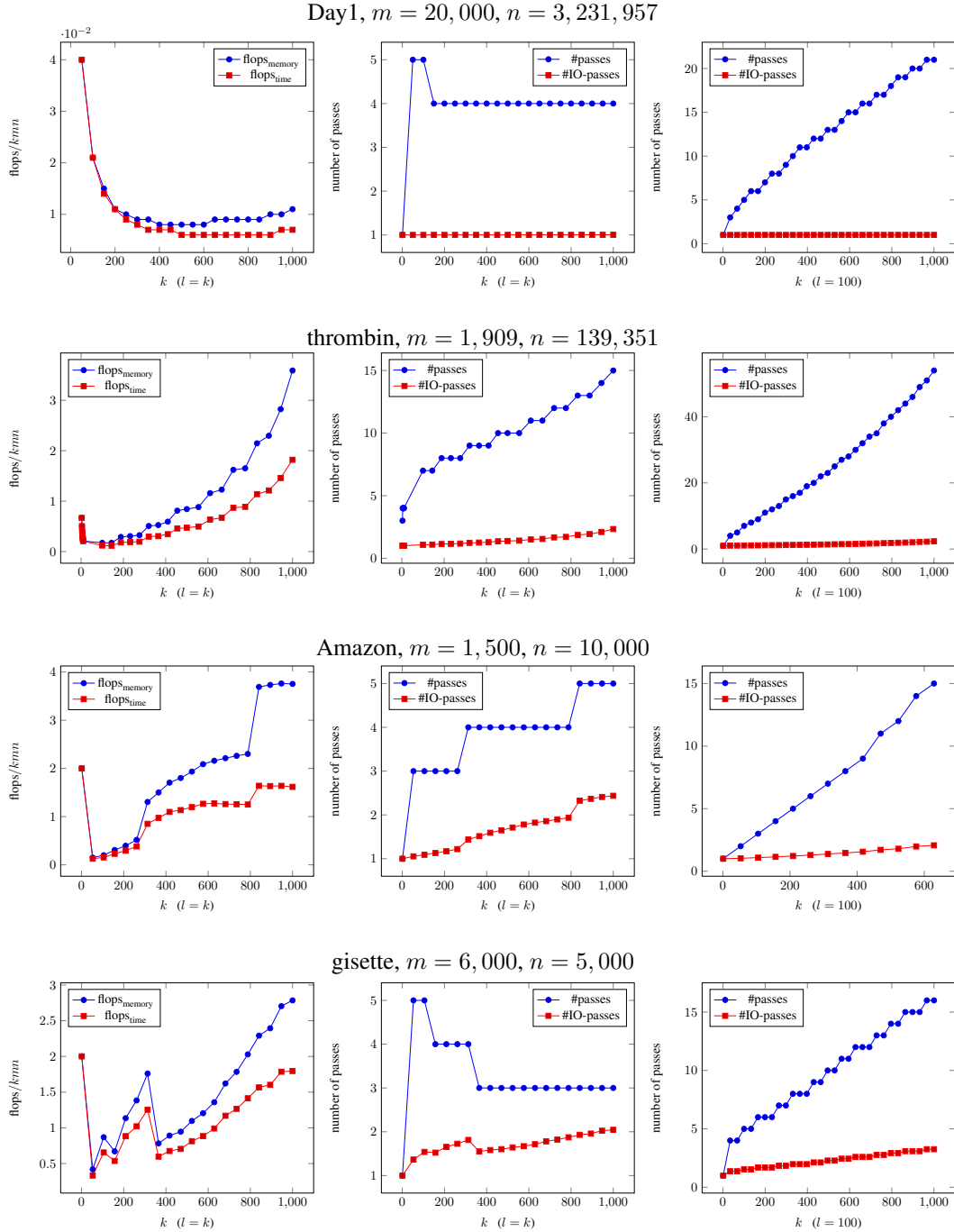

Figure 7: Results of applying the IQRP to several datasets with varying $k$, and $l = k$.

The performance of the IQRP depends on the data. Therefore, the improvements that we observe can also be viewed as an indication that typical datasets are "easy". This appears to suggest that worst case analysis should not be considered as the only criterion for evaluating feature selection algorithms. Comparing the IQRP to the current state-of-the-art randomized algorithms that were reviewed in Section 2.2 we observe that the IQRP is competitive in terms of the number of passes and appears to outperform these algorithms in terms of the number of IO-passes. On the other hand, it may be less accurate.

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
