[Reviews · NeurIPS 2013]

Submitted by Assigned_Reviewer_5

This paper presented a so-called unsupervised feature selection method, which is based on QR decomposition of the data matrix. And the selected features refer to the pivots chosen from QR. The definition here is different from standard feature selection in [A][B]. The authors also discussed several randomized algorithms or sampling methods as related work. Indeed, the problem in this paper relates more closely to matrix factorization rather than unsupervised feature selection.

The complexity of the IQRP is discussed in Section 2.3 and 4, however, the flow of the analysis is not clear and confusing. The authors should provide more details.

One of key advantage of this work is the low complexity. How to select features from very high dimensions in unsupervised setting is an interesting topic. There are a lot of unsupervised feature selection methods, the authors should clarify why choosing features for matrix factorization is important.

The related work discussed in Section 2 are not common unsupervised feature selection. They are neither filter methods, such as mRMR, ReliefF, FCBF nor wrapper based methods, such as RFE using clustering. Moreover, feature selection using random strategies is not very meaningful especially for high dimensional data. Let n = 1000000 and k = 10, it is almost impossible to select k informative features (the probability to choose one of k informative features is 0.00001).

I am not convinced about experimental results. As discussed in [C], redundancy rate is an important criterion for measuring the quality of the selected features. The authors should report this for comparisons. Even for unsupervised feature selection, one can use clustering error or reconstruction error of the data matrix to evaluate the quality for the given number of selected features. From Figure 7, though the proposed algorithm is faster to select k features, I can’t assess the quality of the chosen features. One can randomly choose k features in very short time, but the quality of the chosen features can be very poor. There are other efficient matrix factorization methods [18,19] other than QRP, the authors should compare the qualify of the chosen features given the same amount of run time.

One suggestion is that the authors could use some toy data with known informative features to demonstrate how many informative features can be chosen by the algorithms. Overall the experiment can't demonstrate the validity of the proposed feature selection.

Moreover, the authors should compare with other state-of-the-art unsupervised feature selection methods.

The flow of writing is not easy to follow, the organization can be greatly improved.

[A] Guyon, and Elisseeff, An introduction to variable and feature selection. JMLR., 3:1157–1182, 2003.
[B] Hall, A. Correlation-based Feature Selection for Machine Learning. 1999.
[C] Zhao, Wang, Liu, and Ye, On similarity preserving feature selection. IEEE TKDE, 2013.

Summary: The idea of choosing column for matrix factorization may be interesting. But it is quite different from traditional unsupervised feature selection methods. The authors should clarify the problem setting.

Experiments just reported the timing. However, there is no quality evaluation of the chosen features. Experiments cannot demonstrate the validity of the proposed feature selection methods.

Submitted by Assigned_Reviewer_6

The authors propose a way to accelerate the pivoted QR algorithm by Businger and Golub by selecting multiple columns per iteration instead of one column per iteration as in the classical algorithm. They argue that the selected columns are the same as those selected by the original pivoted QR algorithm, but the algorithm needs much fewer passes through the data.

*Originality*
I am not an expert in numerical linear algebra, but the authors' idea of selecting several features at once seems natural to me. However, as far as I can tell after having superficially browsed related literature, the approach is novel and properly placed in the context of existing work.

*Quality*
The paper seems to be technically sound, but its quality does not fully measure up to NIPS standards in my opinion. The authors do argue in the conclusion that accelerated pivoted QR is competitive with the state of the art, but the claim is not accompanied by empirical results.

*Significance*
The experimental results seem impressive in terms of acceleration wrt classical pivoted QR. However, I may have missed something but it is not clear to me how IQRP compares to other state-of-the-art algorithms. The authors state in Section 6 that IQRP is "competitive" with them in terms of run time but it may be less accurate -- not having more information than this, I am tempted to assume that it is actually not very competitive. Could the authors comment on this?

*Clarity*
The structure of the paper is clear and makes sense, although the writing is a bit dry and lacks flow, and the paper could be more self-contained. E.g., in Section 2.2.2, there is no description whatsoever of "leverage", the authors could at least have given an intuition of what it represents. The experimental section is not well written (e.g. "the real run time on a laptop" is not very specific). If this were a paper submitted to a journal, I would ask for a revision to allow the authors to improve the presentation.

*Typos, comments on style and other details*
- Figure 1 - "indexes" should be "indices".
- Figure captions should start with a capital letter.
- Page 4, line 210, "see, e.g., [21]" should be in parentheses.
- Section 4.1 - use punctuation at the end of equations.
- When citing books, such as [9] or [21], the chapter or the page number should be indicated.
- Figure 1 - After step 1.1 you should add "$\tilde{x}_i = x_i$ for $i \in \{1, \ldots, n\}$", otherwise the \tilde{x}_i are never defined in Figure 2.
- Section 2.3 - the middle paragraph is a second half of the last sentence of the first paragraph, so it should be concatenated to the first one, start with a lowercase letter, and end with a full-stop. In the following paragraph, there is an extra full-stop before "by a large factor".
- Section 5 - "There is still significant savings" should be "There are ...", "Most interesting the observation" should be "Most interesting is the observation", "We change k" should be "We vary k".
- Bibliography, [17] - CUR should be put in curly braces in the bibfile. Same for [13] and [14] and Monte-Carlo.
- Page 8 - "we observe that The IQRP" should be "... the ..."
- "memory efficient" and "time efficient" (resp. "memory optimized", ...) should be "memory-efficient" and "time-efficient" (resp. "memory-optimized", ...)
- Figure 4 - "utilizing" - "using"?
- "Gram Schmidt" - "Gram-Schmidt"
Summary: The authors consider the problem of unsupervised feature selection and propose a data-dependent acceleration of the pivoted QR algorithm by Businger and Golub. The paper is well structured and the contribution may be significant, however experimental comparison with state-of-the-art algorithms is missing, and the writing lacks quality.

Submitted by Assigned_Reviewer_8

This paper proposes improvements to the classical pivoted QR algorithm ("QRP") which lead to time and space advantages useful for feature selection in the large p setting (many variables). The paper is well-written for the most part, and the presentation is clear enough. The algorithm proposed in the paper follows from what might be called low-hanging (or even "obvious") improvements to QRP, in the sense that anyone working with large datasets in practice would begin to think about these and related ideas. But, as in many practical settings, the devil is in the details, and *details matter* particularly in large-scale implementations. This is where the submission shines. The authors have been careful about their CPU and memory bookkeeping at every step, and additionally give a few nice low-level suggestions (e.g., efficient data structures to consider; stability of orthogonalization subroutines is discussed, etc.). On the whole this paper provides a detailed, thorough picture for practitioners working with large datasets, and this is perhaps its most salient selling point: one can go from the paper directly to implementation. I believe this is enough of an advantage to overcome concerns about novelty and the other complaints outlined below.

Other comments:
- The interpolative decomposition is closely related, but it not cited let alone reviewed or compared. This is perhaps a major omission. Yes, the ID is also related to CUR decompositions, which are mentioned, but let's not assume the reader knows this, or by extension, how it ultimately relates to the IQRP algorithm proposed in the paper. See:
- Work from Mark Tygert (NYU)
- N. Halko, P. G. Martinsson, and J. A. Tropp, "Finding
Structure with Randomness: Probabilistic Algorithms for Constructing Approximate Matrix Decompositions," SIAM Review, 2011.

- The ID and other randomized techniques are reasonably efficient. So, if we're going to compare randomized algorithms to IQRP, the efficiency advantage of the latter seems to critically hinge on c being less than k. Why should c be substantially less than k? There is no discussion really exploring this critical, even key, distinction. The experiments suggest it may be true, but the empirical evaluation is limited, and might make it hard to form a concrete opinion as to whether one can expect c << k in a given practical setting going forward. Note that this is apparently what underpins the question as to whether to use IQRP over randomized ID, for instance.

- Experiments: major improvements are needed. For what is mainly an applications/practical paper, the empirical evaluation is a bit underwhelming. Could you show QRP perhaps on the same plots at a minimum? Or comparisons to other, more competitive baselines one would actually consider if faced with making a design decision?

- What is meant by "randomization algorithms can produce better features"? This statement is never really made precise. Also you say "IQRP may be less accurate". In terms of what? It would be interesting to see the accuracy on the regression/classification problems themselves, using the features selected by the various algorithms. Ultimately, this is what we're after, at least for the datasets considered in the paper.

- There are other, related, applications for the pivoted QR decomposition that may help your case. For example, finding nearly dependent columns in forward stepwise regression techniques (and computing related statistics: F-stats, t-stats etc).
Summary: Although the fundamental improvements proposed here are incremental, and the empirical evaluation is lacking for a mostly applied submission, the paper does provide a clear, thorough enough treatment to make the submission a useful, self-contained reference.
Author Feedback

Author rebuttal: We thank the reviewers for constructive comments.
We begin with general comments and then address individual reviewers.

GENERAL COMMENTS

Novelty
-
Some reviewers considered the result as incremental.
We believe this increment to be very significant.
The QRP stood unchallenged (in terms of passes and speed)
for nearly 50 years. As a robust QR it is closely related
to 2 of the 10 most important algorithms of the 20th century
[SIAM news v33 n4].

Our idea is not simply selecting multiple vectors (cf. reviewer-6),
but discovering an efficient method to identify SOME selections
that are PROVABLY identical to QRP selections.
To the best of our knowledge this was not previously
known or even suspected. Being able to run the QRP on big data
may enable, in addition to feature selection, many other
data manipulations that use it as a subroutine.

Comparison with randomized algorithms
-
Most available results on the randomized algorithms we reviewed
are analysis of asymptotic behavior. Implementing them is nontrivial.
We are not aware of experimental evaluation of feature/column selection
with sub kmn performance. Reported experiments (eg [2]) were performed
with a deterministic SVD which takes at least kmn.
The fast leverage algorithms achieve their superior asymptotic complexity
under the condition log n << k, m^2 << kn.
It is unclear at the moment how well they perform for
datasets and k values that do not satisfy these conditions.

With regard to feature quality, since the IQRP produces the same features
as the QRP we didn't think it was essential to report the quality
of the selected features. We provided references that discuss
the quality of the QRP, including the recent [12],
which directly compares the QRP quality with several state-of-the-art
algorithms.

Experimental evaluation
-
We tested many more datasets than what is reported in the paper.
(We also computed several other performance metrics.)
The results are very similar to what is shown,
where we selected the largest well known datasets from UCI.
The constraints we faced were the limited amount of space.

We agree that in the current form the experiments look "underwhelming",
and we will try to significantly improve on that.

REVIEWER-5

Trying to understand the very bad review we checked
references [A,B] provided by the reviewer.
[A] is a classic paper that we know almost by heart. It deals almost
exclusively with the supervised case; there are only 2 paragraphs on
unsupervised selection, and they are consistent with our approach.
(One discusses matrix factorization.)
[B] is a PhD thesis from Waikato that doesn't discuss unsupervised feature
selection at all. The purpose of giving it as a reference is not clear to us.
Perhaps our paper is not clear enough about the distinction between the
supervised and the unsupervised case.

Regarding the need to evaluate feature redundancy as in Reference [C]
given by the reviewer. The features selected by QRP/IQRP are not redundant
by design.

Regarding comments about randomized methods inappropriateness for big data.
A very large and active community, companies such as Yahoo, Google, Amazon,
and grant awarding agencies would all strongly disagree.

REVIEWER-6

The reviewer main concern appears to be the comparison to the current
state-of-the-art. In particular our claim of being competitive.
As explained above we are not aware of implementations of
leverage-based feature selection that use the randomized leverage estimation.
(We are aware of some work in progress..)
Our comparison was based on the asymptotic complexity as reported in
publications. In terms of run time the only potentially
sub kmn algorithm requires fast leverage, with run time of O(mn logn + m^3).
We used a very conservative estimate,
taking the O() notation constant to be 1.
The datasets in our experiments satisfy mn log n + m^3 > kmn
for most k values, while the IQRP runs in sub kmn time.
Therefore in these instances the IQRP is faster.
We believe this justifies our claim of being competitive
in terms of run time.

As for passes we intend to change the text to be less ambiguous.
The IQRP is competitive in terms of the number of passes,
and typically outperforms the randomized algorithms in terms of effective
passes. Here again the competition is with the fast leverage algorithms.
In 2.2.2 we pointed out that one may need to run the randomized leverage
procedure several times. Running it once requires 4 passes,
running it twice requires 6 passes.
The results reported in Fig.7 are competitive,
since only the thrombin takes more passes, and only for high k values.
In our experiments the number of effective passes of the IQRP
was almost always below 2. This is better than the number of passes
of the randomized algorithms we discussed.

We will define leverage and accuracy, and remove laptop runtime.


REVIEWER-8

Yes, the technique detailed in Halko et.al. for computing the QRP
using ID should have been discussed. It is a major omission but it
will be easily fixed.
In a nutshell, their technique may be faster than ours when
the matrix rank is approximately k. Otherwise their row extraction
step will introduce unacceptable inaccuracies, which led Halko et.al
to recommend using an alternative O(kmn) technique. See their Remark 6.1.
(Our experience indicates that in typical datasets the above condition
doesn't hold. It would imply that the entire data can be very accurately
estimated by the k selected features.)

With regard to run time guarantees we will add a proof that
the IQRP is no slower than the QRP when l~k. Anything above that
is based on experimental data.

We intend to improve the presentation, experiments, and properly define
accuracy and leverage.